# Bright Morning Lighting Enhancing Parasympathetic Activity at Night: A Pilot Study on Elderly Female Patients with Dementia without a Pacemaker

**DOI:** 10.3390/healthcare11060793

**Published:** 2023-03-08

**Authors:** Chuen-Ru Liu, Terry B. J. Kuo, Jwo-Huei Jou, Chun-Ting Lai Lai, Yu-Kai Chang, Yiing Mei Liou

**Affiliations:** 1Taipei City Hospital, Taipei 110, Taiwan; iqlulu@mail2000.com.tw; 2Institute of Brain Science, Sleep Research Center, National Yang Ming Chiao Tung University, Taipei 112, Taiwan; 3Department of Materials Science and Engineering, National Tsing Hua University, Hsinchu 30013, Taiwan; 4Department of Physical Education and Sport Sciences, National Taiwan Normal University, Taipei 106, Taiwan; 5Institute of Community Health Care, College of Nursing, National Yang Ming Chiao Tung University, Taipei 112, Taiwan

**Keywords:** light therapy, dementia, heart rate variability, cognitive function, parasympathetic, sympathetic

## Abstract

Exposure to bright morning light (BML) entrains the master circadian clock, modulates physiological circadian rhythms, and reduces sleep–wake disturbances. However, its impact on the autonomic nervous system at night remains unclear. Here, we investigated the effects of BML exposure on parasympathetic nervous system (PSNS) and sympathetic nervous system (SNS) activity at night in elderly women. This nonrandomized controlled pilot study included female participants aged ≥ 60 years who were diagnosed with a type of dementia or cognitive disorder, excluding individuals with pacemakers. The treatment group was exposed to 2500 lx of BML, whereas the control group was exposed to 200 lx of general lighting. We measured heart rate variability to quantify ANS activity. The treatment group displayed significant increases in high-frequency (HF) power (Roy’s largest root = 1.62; *p* < 0.001) and nonsignificant decreases in normalized low-frequency (LF%) power. The corresponding nonsignificant decreases in the low-frequency/high-frequency (LF/HF) ratio and cognitive function were correlated with PSNS activity (Roy’s largest root = 1.41; *p* < 0.001), which improved severe dementia. BML exposure reduced SNS activity and enhanced PSNS activity at night in female participants, which improved cognitive function. Thus, BML therapy may be a useful clinical tool for alleviating cognitive decline.

## 1. Introduction

In this demographically aging world, approximately 55 million people have dementia worldwide, which is expected to increase to 78 million by 2030 and 139 million by 2050. Dementia’s estimated prevalence and growth rates are increasing, leading to increased medical expenditure of hundreds of billions of USDs [1]. Dementia-related health problems have become a prioritized topic at national and community levels [1]. Notably, more women are affected by dementia than men. Overall, twice as many women have dementia compared to Alzheimer’s disease [2]. Menopause affects the secretion of sex hormones in women, such as progesterone, testosterone, and estrogen levels, which increases the risk of dementia in menopausal women. Based on estrogen and progestin medroxyprogesterone acetate, researchers have proposed the “healthy cell hypothesis” in which estradiol exerts neuroprotective effects [2,3]. Dementia is a neurodegenerative process that affects memory, social behavior, activities of daily living, and sleep; reduced nocturnal parasympathetic activity may represent early predictive factors for progression to dementia [4]. Dementia is often characterized by autonomic dysfunction, particularly reduced parasympathetic nervous system (PSNS) activity. Furthermore, greater sympathetic nervous system (SNS) activity at night is associated with poorer cognition and sleep quality [5,6]. Cognitive disorders in individuals with dementia lead to increased SNS activity; decreased PSNS activity appears to be associated with worse performance in cognitive domains, which may be associated with impaired autonomic regulation and self-regulatory neural circuits [4,7].

The principal regulators of the sleep–wake cycle and the control of autonomic activity are located in the suprachiasmatic nucleus (SCN), predominantly including the neuroendocrine and hypothalamic pathways that balance the mechanisms of nerve conduction [8,9]. The SNS is similar to the gas pedal of an automobile, energizing the body to cope with external challenges; however, the PSNS acts like the braking system, decelerating the metabolic rate to facilitate the body to enter into a quality sleep mode. The PSNS maintains a default state, contributing to baseline repair and nutrient storage functions. In contrast, the SNS includes neurological and humoral elements that facilitate real-time reactions to dyshomeostasis. The activity of the ANS, consisting of the SNS and PSNS, reflects normalized low-frequency (LF%) and high-frequency (HF) power in heart rate variability (HRV) data. The LF/HF ratio quantifies the sympathovagal balance to reflect the modulations of SNS activity [10,11].

BML exposure is a noninvasive, simple, easy-to-operate, and cost-effective therapy with limited side effects for individuals with age-related sleep disturbances [12]. These effects of light are mediated by intrinsically photosensitive retinal ganglion cells (ipRGCs). IpRGCs involve a pacemaker-independent input to the SCN, which is the master circadian clock in the human body that is localized in the hypothalamus over the optic chiasm [13]. The SCN is vulnerable to aging characterized by a decreased sensitivity of the eyes to light and diminished transmission of light by the lenses, caused by a reduced pupil diameter and yellowing of the lenses. Consequently, these factors hinder light entry, which is necessary for rhythm synchronization, thus explaining the worsening of circadian disruption in older people with insufficient exposure to daylight during the day [14]. Light therapy can reduce night-time awakenings, enhance sleep quality [15], and advance the circadian rhythm phase, thus reducing the phase delay and night-time awakenings; eventually, it reduces sleep disturbances [16]. Exposure to BML induces advances in sleep onset, delays sleep offset [17], and improves cognitive function [13,18] in older adults with dementia. Therefore, sleep quality can be used as an index of PSNS activation and SNS deactivation at night [6,10,11,19].

The ANS is strongly associated with the central autonomic network. The bidirectional communication between these systems is the crossover of the connection between the brain and body, where the brainstem structures supposedly generate output to the heart [20]. HRV is the complex beat-to-beat variation in the heart rate produced by the interactions of sympathetic and parasympathetic (vagal) neural activity at the sinus node [21]. HRV is a valid noninvasive measure for detecting awakenings during sleep and has been applied to understand the effects of insomnia, sleep disorders, cardiorespiratory function, immune function, cognitive function, and neurodegenerative diseases in individuals with dementia [6,7,9,22]. The relationships among HRV parameters provide additional insights into the mechanisms underlying autonomic homeostasis [4,6]. According to ANS studies conducted at night, washout light exposure reduces the effect and cognitive function. However, the impact of BML exposure on the ANS at night is unclear, particularly in elderly female patients with dementia. There are insufficient data on the effects of BML exposure on the brain.

We hypothesized that exposure to BML would exert beneficial effects on autonomic regulation at night, with an increase in PSNS activity and a decrease in SNS activity. We aimed to perform longitudinal experiments using noninvasive recording methods. Furthermore, we intended to explore the relationship between BML exposure and cognitive function.

## 2. Materials and Methods

### 2.1. Study Design

We conducted a nonrandomized controlled pilot study using a convenience sampling design. A single-blind approach was used in this study. The research protocol (YM107120F) was reviewed and approved by the Institutional Review Board. The participants and their caregivers signed the written informed consent form before enrollment.

BML (2500 lx) and general light were used for the treatment group and control group, respectively, in a parallel intervention study design. The study protocol had a duration of 5 weeks. We assessed the outcome measures at the end of the baseline and after 5 weeks of the intervention, thus collecting data at the following two time points: baseline and week 5 (posttest 1).

### 2.2. Participants

The inclusion criteria were as follows: (1) diagnosis of any type of dementia or cognitive disorder according to the Diagnostic and Statistical Manual of Mental Disorders, fifth edition [23]; (2) age ≥ 60 years; (3) female sex; (4) consent to participate in the study (both participants and their caregivers); and (5) willingness to be monitored using an HRV instrument at the end of the study. The exclusion criteria were as follows: (1) conditions that caused adverse reactions to light, such as systemic lupus erythematosus, epilepsy, retinal detachment, or macular degeneration, or (2) a pacemaker and a medical history of arrhythmia [24], according to the reports of caregivers and nursing home staff.

Among 60 eligible participants, 19 did not meet the inclusion criteria, and 14 dropped out of the study. Accordingly, we enrolled 27 patients consecutively allocated to the treatment (*n* = 16) and control *(n* = 11) groups. Ultimately, only 22 patients completed the study (Figure 1).

### 2.3. Intervention Group

The treatment group received BML therapy with horizontal illuminations of 2500 lx delivered by a panel light (Figure 2) in a designated room from 9:00 AM to 10:00 AM for 5 weeks (5 days/week). Thus, they received a total of 25 h of BML therapy in a group setting. The intervention and screening procedures have been previously reported [17,18].

### 2.4. Instruments and Outcome Measures

We determined the primary outcome measures, including the mean values of the trial variables, HF power, LF power, and the LF/HF ratio, from the HRV data. We used the PSNS activity indicator and HF component determined from the HRV data. The SNS activity indicator and LF% component were determined from the HRV data. The sympathovagal balance indicator and LF/HF ratio were derived from the HRV measures. Data were analyzed at baseline and posttest 1 between April 2019 and December 2019. The data collectors were blinded to the group allocation. We determined the corresponding SNS outcomes using LF% power and the sympathovagal balance indicator using the LF/HF ratio; the PSNS data were obtained using HF power. Cognitive function was the secondary outcome measure assessed by the researchers who directly conducted the Mini-Mental State Examination (MMSE). They explored the relationships of cognitive function with HF power, LF% power, and the LF/HF ratio.

#### Validity and Reliability of HRV Data

We assessed one component each in the LF range (0.04–0.15 Hz) and HF range (0.15–0.40 Hz). The LF power was normalized to HF power ([HF/(HF + LF) × 100]) to remove the effects of very-low-frequency (VLF) bands and to detect the effects of SNS activity on HRV. We used the LF/HF ratio to quantify the sympathovagal balance and reflect the modulations of SNS activity. The LF/HF ratio was significantly negatively correlated with the delta power measured using electroencephalography (0.5–4.0 Hz) during quiet sleep, whereas cardiac sympathetic regulation was negatively correlated with the depth of sleep [25]. Detailed procedures for the analysis of HRV have been previously reported [26]. Pacemaker use and arrhythmia were associated with increased heart rate fragmentation, which reduced the HRV accuracy [27].

The HRV device (WG-103A, Taipei, Taiwan) was placed over the sinus node to enable continuous long-term recording during 1–7 h of lying in bed overnight and for ≥1 day at baseline and posttest 1 using the KY laboratory software package (http://xenon2.ym.edu.tw/hrv/ (accessed on 1 December 2019)). We adjusted the HRV parameters for interference factors, system settings, and statistical settings, excluding extreme values. The real-time signal was transmitted to a smartphone using Xenon Bluetooth Low Energy. Specifically, HRV was recorded as a precordial electrocardiograph. HRV exhibits higher sensitivity and lower specificity than polysomnography [28,29]. HRV measurements were continuously performed between 7 PM (19:00) and 7 AM (07:00).

### 2.5. Statistical Analyses

We used repeated measurement generalized linear models (GLMs) to analyze the effects of BML exposure on the HRV parameters at night [30,31]. The GLMs included results collected over 7 consecutive hours. We compared the efficacy of BML therapy to that of general light therapy and the time point (baseline and posttest 1), whereas the “interaction” (group × time point) was used to calculate the statistical significance. The analyses were performed using SPSS 24 software, and statistical significance was set at *p* < 0.05.

Therefore, a significant interaction effect indicated a significant between-group (treatment vs. control groups) difference in the changes over 1–7 h of lying in bed overnight (posttest 1 and baseline). Improvements in MMSE scores over time in both groups were analyzed using generalized estimating equations with an exchangeable working correlation matrix. A robust standard error was used to calculate statistical significance. Finally, we used the MMSE to determine the cognitive severity of dementia. This helped us explore the relationships between cognitive function and HRV parameters. Scores ≥ 21, 11–20, and 0–10 indicated the presence of mild, moderate, and severe dementia, respectively [32].

## 3. Results

### 3.1. Demographic and Clinical Characteristics

Figure 1 shows the study flowchart. Thirty-four participants completed the baseline and posttest 1 assessments. The primary pre- and posttest analyses were based on the intention-to-treat (ITT) sample. The retention rates were 88% and 73% in the treatment and control groups, respectively. Table 1 summarizes the between-group comparisons of the demographic characteristics using chi-squared tests. There were no differences in participants’ baseline and posttest characteristics between the groups. Furthermore, Mann–Whitney U tests did not detect differences in age, education level, marital status, dementia severity, dementia type, recruitment source, or medication use between groups. Therefore, an independent sample *t* test was used to assess HF power, LF power, the LF/HF ratio, and the MMSE score at baseline. No significant differences were observed between groups, indicating the homogeneity of the participants’ distribution. The patients’ ages were 83.7 (SD, 7.3) and 82.3 (SD, 4.5) years; MMSE scores were 14.3 (SD, 6.7) and 16.7 (SD, 8.4) in the treatment and control groups, respectively. This study could not achieve random sampling due to several practical issues, including difficulties in transporting all the participants from their communities or institution to the light laboratory and interference from some of the participants’ family members who insisted on accompanying them during the experiment.

### 3.2. Primary Outcomes

#### 3.2.1. Effects of BML on PSNS Activity

The present study used the PSNS activity indicator (the HF component determined from HRV data) to assess the effects of BML exposure. We analyzed the main effect of the group × time interaction on HF power; HF power significantly improved. Specifically, we identified significant differences in HF power in the treatment group while lying in bed for 1–7 h at baseline compared with posttest 1 (Roy’s largest root = 1.62; *p* < 0.001) (Figure 3a and Table 2).

The BML group demonstrated a sustained peak in HF power while lying in bed for 2–4 h. Specifically, the BML group had an increased peak in HF power at 2 h at posttest 1 (Figure 3b). Therefore, the BML group exhibited higher PSNS activity than the control group, resulting in stable and regular circadian rhythms.

#### 3.2.2. Effects of BML on SNS Activity

The present study used the SNS activity indicator (the LF% component of HRV) to assess the effects of BML exposure. We observed a significant main effect of the treatment on the LF% power (Roy’s largest root = 0.25; *p* > 0.05). The treatment group demonstrated a decrease in LF% power compared with the control group. The LF% values at posttest 1 were lower than those at baseline; however, the difference was statistically nonsignificant (Figure 4a and Table 2). Specifically, the treatment group demonstrated a significant decrease in LF% power while lying in bed for 2–3 h, resulting in a U-shaped relationship curve between LF% power and time at posttest 1 (Figure 4b).

#### 3.2.3. Effects of BML on Sympathovagal Balance

We used the sympathovagal balance indicator (the LF/HF ratio derived from HRV measures) to assess the effects of BML exposure. We analyzed the effect of the group × time interaction on the LF/HF ratio. We observed a nonsignificant decrease in the LF/HF ratio of the treatment group (Roy’s largest root = 0.24; *p* > 0.05).

The treatment group demonstrated a larger mean decrease than the control group at subsequent time points relative to baseline (Figure 5a and Table 2). Specifically, individuals in the treatment group demonstrated a significant decrease in the LF/HF ratio while lying in bed for 3–4 h, generating a U-shaped relationship curve between the LF/HF ratio and time at posttest 1 (Figure 5b).

### 3.3. Secondary Outcomes

#### Effect of HF, LF%, and LF/HF on Cognitive Function

The treatment group showed significant improvement in the MMSE score (Wald’s test = 4.4, *p* < 0.001) from baseline to posttest 1 compared with the control group. A main effect on cognitive function was observed. A significant change in the slope indicated improved cognitive function (Figure 6a).

The treatment group demonstrated a significant correlation between cognitive function and HF power (Roy’s largest root = 1.41; *p* < 0.001). BML exposure enhanced PSNS activity at night, which was more effective for severe dementia patients than for moderate or mild dementia patients. Patients with severe dementia in the BML and general lighting groups had mean PSNS activity scores of 5.02 and 4.55, respectively. Those with moderate dementia in the BML and general lighting groups had mean scores of 5.14 and 4.80, respectively. Those with mild dementia in the BML and general lighting groups had mean scores of 3.93 and 3.36, respectively (Figure 6b). However, the differences in LF% power and the LF/HF ratio did not reach statistical significance. Furthermore, the activation of the PSNS in older adults with dementia was associated with cognitive function at night in females. This can be attributed to a significant correlation between PSNS activation and cognitive function (Table 2).

## 4. Discussion

This pilot study demonstrated that BML exposure significantly enhanced PSNS activity, reduced SNS activity, and improved cognitive function, enabling stable and regular circadian rhythms. BML exposure significantly enhanced PSNS activity, with a peak at posttest 1. Furthermore, night-time PSNS activity was associated with cognitive function. The BML intervention increased the peak PSNS activity while the participants were lying in bed for 2 h. In contrast, the decrease in SNS activity while the participants were lying in bed for 2–3 h resulted in a U-shaped relationship curve between SNS activity and time. The reduction in sympathovagal balance while the participants were lying in bed for 3–4 h resulted in a U-shaped relationship curve between sympathovagal balance and time.

BML exposure significantly enhanced PSNS activity and reduced SNS activity at night. Therefore, BML exposure can suppress melatonin secretion and induce serotonin secretion, which reduces the phase delay and night awakening, thereby reducing sleep disturbance. Relevant studies have reported that BML improves sleep disturbances, circadian rhythms, cognitive function, and behavioral and psychological symptoms of dementia [12,15,16,17,18]. Therefore, we recommend BML for driving sleep propensity. PSNS activity was correlated with cognitive function; however, the present study included fewer participants. This necessitates additional research with a larger cohort to confirm our findings.

### 4.1. BML Exposure Enabled Stable Circadian Rhythm and the Postponement of Dementia Disease Progression

A previous study reported that morning light cures, whereas night light kills [34]. Exposure to light at night (LAN), especially intensive, blue-enriched white light, greatly suppresses melatonin secretion and disrupts sleep homeostasis, causing sleep disturbance, metabolic syndrome, autonomic circadian rhythm disruptions, dyshomeostasis, and neuropsychiatric symptoms [35,36]. Therefore, exposure to BML and darkness after dusk benefits patients with sleep disturbances. The resulting synchronization of the circadian rhythm represents an effective intervention for individuals with neurodegenerative diseases [13].

The effects of light exposure are mediated by ipRGC pathways involving a pacemaker-independent input to the SCN. The SCN is the biological basis for circadian rhythm (rest–activity, sleep–wake cycles, master circadian clock, autonomic activities, body temperature, and melatonin excretion rhythm). However, it becomes vulnerable in the aging process, particularly in people with dementia [8,9,37], while longer sleep deprivation affects the circadian clock, which is less susceptible to light [35].

BML increases the amplitude of SCN signals [12,15,38], indicating that a strong central clock induces periods of deep sleep and is efficacious against neurodegenerative diseases [13,35]. The circadian rhythm plays a fundamental role in regulating biological functions, including waking time and hormone secretion. Furthermore, clock genes play important roles in circadian regulation. However, a disruption in the expression of these genes underlies the pathological mechanisms of metabolic diseases, cancer, and neurodegenerative diseases [13,16,17,39]). BML activates molecules and cells in the body, such as immune cells, inflammatory mediators, and bone marrow-derived stem cells. These cells help preserve neuronal functions by releasing nerve growth or brain-derived neurotrophic factors. These neurotrophic factors alter neurotransmission and disrupt the levels of neurotrophic factors, causing the behavioral and psychological symptoms of dementia [13].

PSNS activity peaked while the participants were lying in bed from 2 AM at baseline, whereas maximal PSNS activity was observed in healthy participants at 2 AM [36]. BML exposure caused PSNS activity to peak, potentially reducing the phase delay. Furthermore, PSNS activity peaked while the participants were lying in bed from 1 to 2 AM, whereas maximal PSNS activity was observed in healthy participants at 2 AM [40]. The body reaches its lowest temperature between 4 AM and 5 AM; BML exposure after this time will adjust the biological clock to advance sleep onset [30]. BML therapy is the recommended treatment to induce the resynchronization of circadian rhythms, reduce sleep disturbance, and mitigate the development of neurodegenerative diseases [29]. For BML therapy to be effective, a threshold of ≥2500 lux light intensity is suggested for ≥60 min/day in the morning for ≥4 weeks [13,17]. Meta-analyses of BML have shown it to be effective in improving sleep disturbances and circadian rhythms [15,16,17]. In contrast, BML exposure washout compromises the treatment outcomes [41]; therefore, continuous BML is helpful and necessary.

The present study’s findings were consistent with previously reported outcomes. BML exposure resulted in improvements, such as the induction of stable and regular circadian rhythms, in the treatment group compared with the control group.

### 4.2. BML Exposure Enhanced PSNS Activity to a Maximum at Night

The effects of BML might include the alleviation of autonomic nervous load, regulation of circadian rhythms, and integration of neuroendocrine systems, the last of which is indicated by an increase in gamma-aminobutyric acid (GABA) levels and a decrease in glutamate levels [42]. GABA is an inhibitory neurotransmitter that suppresses arousal pathways from the brainstem to higher brain centers and initiates the release of relaxation-related neurotransmitters (for example, blocking glutamate release and adenosine release inhibitors). GABA is a key regulator of the vagus nerve and is involved in the PSNS. PSNS is largely mediated by the vagus nerve and can “turn off” in response to stressors, which can increase cerebral hyperperfusion [42,43]. For example, decreased heart rate and blood pressure at night may positively correlate with PSNS activity [44].

BML leads to promising outcomes and exerts immediate positive effects on mood, energy level, and blood oxygen saturation. In addition, it decreases the mean heart rate in older adults with dementia. Biological responses are physiological indicators of a relaxation response [30]. BML exposure immediately triggers PSNS activity, even at night.

Kong et al. [4] explored HRV during sleep in older people at risk of dementia. Participants with amnestic mild cognitive impairment demonstrated impaired parasympathetic regulation, particularly during nonrapid eye movement sleep. Mental stress induces the activation of the amygdala through the central autonomic system. Efferent projections increase the activation of the SNS and lead to a proinflammatory state that initiates and promotes atherosclerosis, in which cerebral hypoperfusion affects autonomic, mood, and cognitive regulatory cerebral sites [3]. Furthermore, SNS activity may represent the efferent arm of inflammatory responses, immune responses, and cardiovascular risk [9]. The vagus nerve acts as a “brake” for the SNS when the PSNS deficit leads to an unrestrained SNS [44]. In the literature, a washout BML intervention was proposed to reduce vagus nerve actions and trigger boosts in the SNS.

BML exposure enhanced PSNS activity at night, whereas the washout BML intervention boosted SNS activity. This feature is key to postponing the deterioration associated with dementia and warrants further research.

### 4.3. BML Therapy Is More Effective for Severe Dementia-Enhanced PSNS Activity

BML-enhanced night-time PSNS activity was correlated with cognitive function, and the severity of dementia influenced the elevation of HF power at night. Therefore, BML therapy is more effective for severe dementia than for moderate and mild dementia.

The present study’s findings were consistent with previously reported systematic outcomes, in which increased SNS activity and decreased PSNS activity appeared to be associated with worse performance in cognitive domains. PSNS activity is associated with the optimal functioning of the prefrontal–subcortical inhibitory circuits that sustain quick and flexible responses to environmental demands [45]. It has been associated with greater cortical thickness, a larger amygdala volume in prefrontal regions, and better performance in attention and executive function tests [44,46]. In contrast, hyperarousal states have been associated with reduced activation of the brain regions that underlie cognitive control in the ventromedial prefrontal cortex [20]. Enhancing PSNS activation at night can be an index of sleep quality [4,8,9,17].

Memory consolidation is primarily attributed to rapid eye movement sleep. The ability to form and retain nonemotional fact-based memories is associated with the presence and integrity of slow-wave sleep (SWS) [47,48]. Disrupted sleep and decreased SWS impair frontal lobe restoration, whereas sleep quality treatments that facilitate amyloid-beta clearance and SWS for memory processing focus on memory encoding and consolidation [47,48,49]. Cognitive function was correlated with PSNS activity in the present study, consistent with previous findings [4,7]. Bright light exposure improves attentiveness and cognitive function [12,16,18]. Sleep is crucial in memory consolidation [47,50]. In this pilot study, BML exposure improved cognitive function and PSNS activity. Additionally, cognitive function was correlated with PSNS activity. Hence, BML exposure may improve cognitive function by increasing PSNS activity (an indirect effect on cognitive function). Future research should verify whether the effect of BML exposure on cognition is mediated by PSNS activity.

Nighttime PSNS activity is the key predictive factor for sleep-dependent cognitive function because it mitigates deterioration in cognitive function. Therefore, reductions in night-time PSNS activity may be an early marker of cognitive decline in older individuals with dementia.

### 4.4. Limitations

This study had some limitations. First, this pilot study utilized a convenience sampling design, which affected generalizability beyond the present population. Second, the sample size was relatively small, reducing the statistical power and increasing the margin of error, potentially affecting the results. Third, the inclusion criterion for participants was women; therefore, the present study’s findings may not be generalizable to men. Fourth, we did not control for the factors affecting HRV, such as hypertension, diabetes mellitus, or dementia type. Fifth, we could not control other aspects, such as the timing of metabolic disease, medication use, sleep–wake changes, or age.

## 5. Conclusions

In this study, BML exposure enhanced PSNS activity and correlated with cognitive function, which is effective for treating severe dementia. Furthermore, the light washout intervention significantly boosted SNS activity at night in females. This intervention is a noninvasive, simple, and easy-to-operate therapy for older individuals with dementia. It should be continuously applied to maintain effectiveness.

## Figures and Tables

**Figure 1 healthcare-11-00793-f001:**
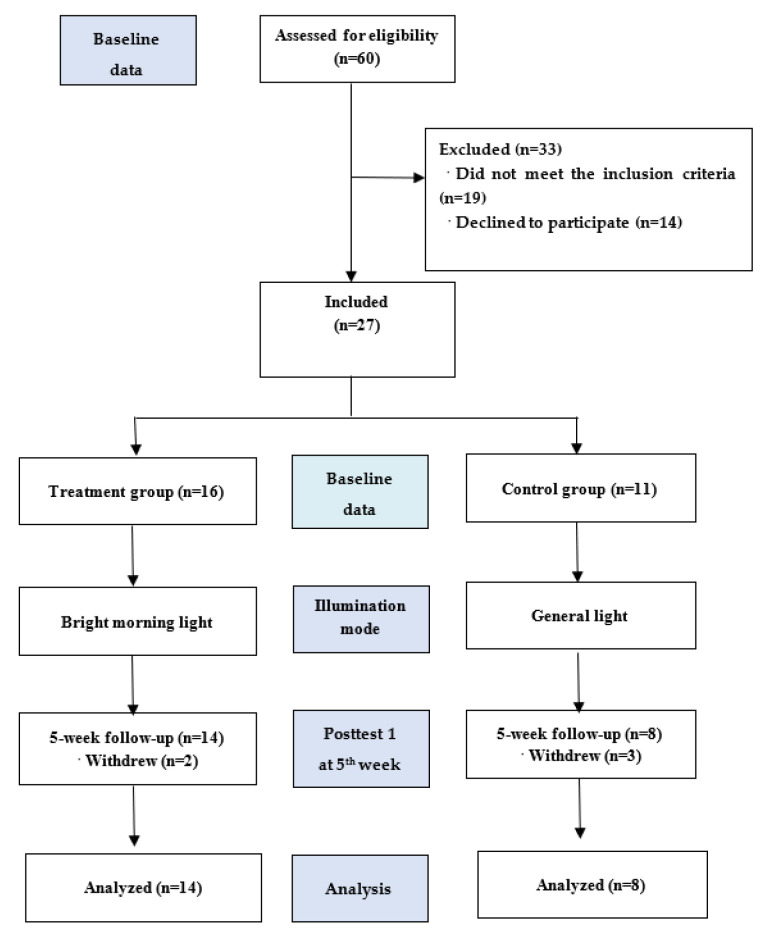
Flow chart of participant recruitment.

**Figure 2 healthcare-11-00793-f002:**
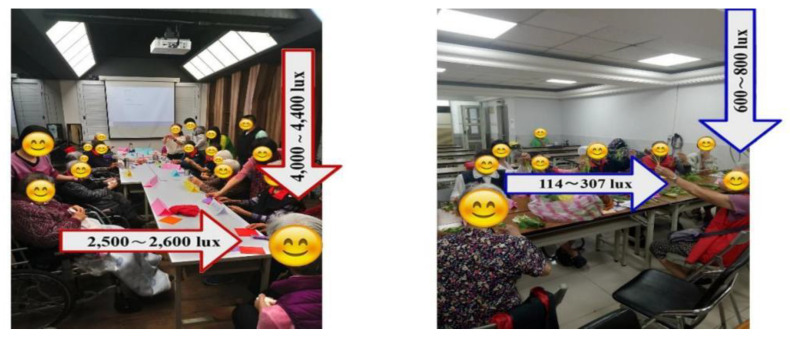
Representative images of the following exposure conditions: bright morning light (2500 lx) or general light (200 lx).

**Figure 3 healthcare-11-00793-f003:**
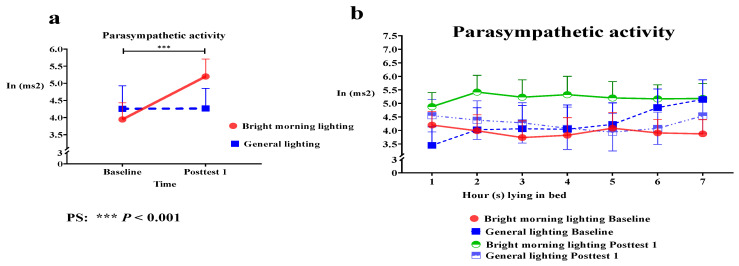
(**a**) Changes in parasympathetic nervous system (PSNS) activity in both the treatment and control groups while lying in bed for 7 h. PSNS activity shows a significantly different high-frequency power (HF) component from baseline to posttest 1 in the treatment group compared with the control group (*p* < 0.001). (**b**) The BML group demonstrated a sustained peak in HF power while lying in bed for 2–4 h. Specifically, the BML group had an increased peak in HF power at 2 h at posttest 1.

**Figure 4 healthcare-11-00793-f004:**
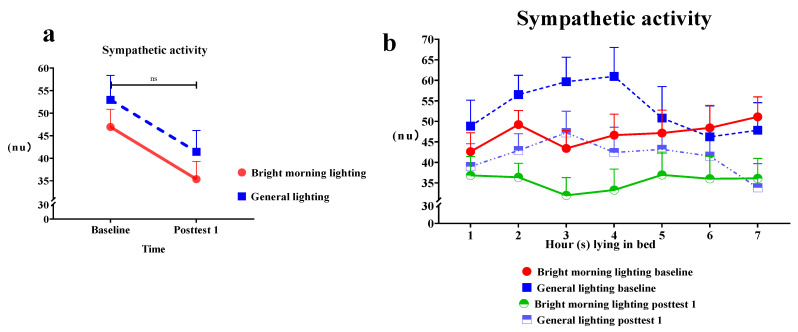
(**a**) Changes in sympathetic nervous system (SNS) activity in the treatment and control groups while lying in bed for 7 h. The treatment group exhibited a nonsignificant decrease (*p* > 0.05). (**b**) The BML group exhibited a significant decrease in LF% power while lying in bed for 2–3 h. A U-shaped curve was exhibited at posttest 1.

**Figure 5 healthcare-11-00793-f005:**
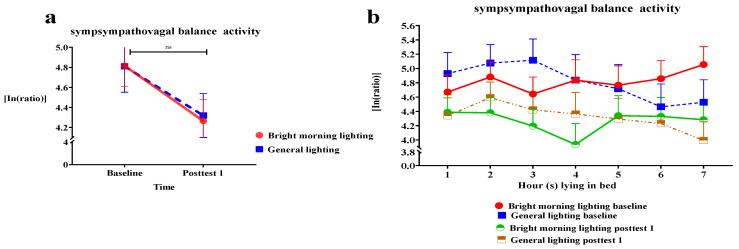
(**a**) A nonsignificant decrease in the low-frequency/high-frequency (LF/HF) ratio of the treatment group (*p* > 0.05). (**b**) The treatment group demonstrated a significant decrease in the low-frequency/high-frequency (LF/HF) ratio while lying in bed for 3–4 h, resulting in a U-shaped curve.

**Figure 6 healthcare-11-00793-f006:**
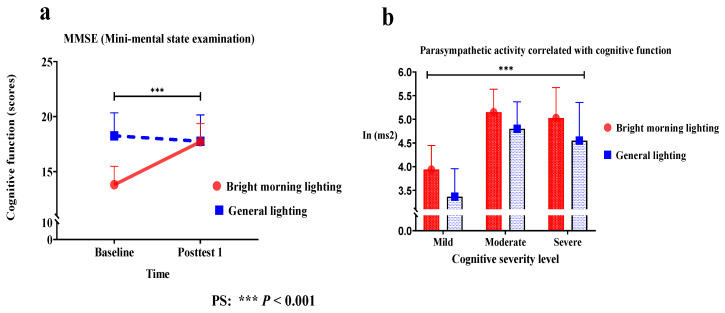
(**a**) Exposure to bright morning light (BML) significantly improved MMSE scores (Wald’s test = 4.4, *p* < 0.001) from baseline to posttest 1 compared with exposure to general lighting. A main effect of lighting type on cognitive function was observed. A significant change in the slope indicated improved cognitive function. (**b**) The treatment group demonstrated a significant association between cognitive function and high-frequency (HF) power (Roy’s largest root = 1.41; *p* < 0.001). Bright morning light (BML) exposure enhances parasympathetic nervous system (PSNS) activity at night, which is more effective for severe dementia than for moderate and mild dementia.

**Table 1 healthcare-11-00793-t001:** Participants’ characteristics at baseline and posttest 1.

Characteristic	Time	Treatment Group (*n* = 16)		Control Group (*n* = 11)		Chi-Square Test		Time	Treatment Group *(n* = 14)		Control Group (*n* = 8)		Chi-SquareTest	
		*n*	(%)	*n*	(%)	χ^2^	*p*		*n*	(%)	*n*	(%)	χ^2^	*p*
Educational level	Baseline					2.52	0.47	Posttest 1					0.55	0.90
1. Literate		6	(37.5)	2	(18.2)				5	(35.7)	2	(25.0)		
2. Elementary School (≤6 years)		5	(31.3)	3	(27.2)				4	(28.6)	2	(25.0)		
3. Junior high school (7–9 years)		3	(18.7)	2	(18.2)				3	(21.4)	2	(25.0)		
4. Senior high school or college (10–12 years)		2	(12.5)	4	(36.4)				2	(14.3)	2	(25.0)		
Marital status						2.18	0.33						1.62	0.43
1. Unmarried		2	(12.5)	0	(0.0)				2	(14.3)	0	(0.0)		
2. Married		3	(18.8)	4	(36.4)				3	(21.4)	3	(37.5)		
3. Widowed		11	(68.7)	7	(63.6)				9	(64.3)	5	(62.5)		
Dementia type						6.80	0.079						0.73	0.69
1. Alzheimer		14	(87.4)	10	(90.9)				12	(85.8)	7	(87.5)		
2. Vascular dementia		1	(6.3)	1	(9.1)				1	(7.1)	1	(12.5)		
3. Frontotemporal lobe		1	(6.3)	0	(0.0)				1	(7.1)	0	(0.0)		
Severity of dementia						1.08	0.58						0.10	0.94
1. Mild		3	(18.8)	4	(36.4)				6	(42.8)	4	(50.0)		
2. Moderate		8	(50.0)	4	(36.4)				4	(28.6)	2	(25.0)		
3. Severe		5	(31.2)	3	(27.2)				4	(28.6)	2	(25.2)		
Source						1.16	0.28						2.16	0.14
1. Community		10	(62.5)	9	(81.8)				8	(57.1)	8	(87.5)		
2. Nursing home		6	(37.5)	2	(18.2)				6	(42.9)	1	(12.5)		

**Table 2 healthcare-11-00793-t002:** GLM analysis of the longitudinal changes in HRV and MMSE scores.

Step	HF ^a^ Mean/SE			Roy’s Largest Root ^h^/t ^i^	LF% ^b^ Mean/SE			Roy’s Largest Root ^h^/t ^i^	LF/HF ^c^ Mean/SE			Roy’s Largest root ^h^/t ^i^	MMSE Score ^d^ Mean/SE			Wald χ^2 j^ /t ^i^
	Exp ^e^	Con ^f^	Diff ^g^	*p* Value	Exp ^e^	Con ^f^	Diff ^g^	*p* Value	Exp ^e^	Con ^f^	Diff ^g^	*p* Value	Exp ^e^	Con ^f^	Diff ^g^	*p* Value
Week 0 ^k^	3.9 0.48	4.2 0.67	−0.3	0.54 ^i^	47.0 3.91	52.9 5.40	−5.9	−0.98 ^i^	4.8 0.20	4.8 0.25	0.0	−0.87 ^i^	13.80 1.66	18.25 2.09	−4.45	1.38 ^i^
				0.59				0.33				0.38				0.41
Week 5 ^l^	5.2 0.50	4.2 0.58	1.0		35.3 3.91	41.4 4.71	−6.1		4.2 0.21	4.3 0.21	−0.1		17.71 1.65	17.75 2.39	−0.04	
Group ^m^	4.7 0.31	4.2 0.38	0.5	1.49 ^h^	39.3 3.78	45.1 4.36	−5.1	2.9 ^h^	4.5 0.13	4.6 0.13	−0.1	1.63 ^h^	16.97	18.47	−1.50	−4.45 ^j^
				0.21				0.25				0.16				0.09
Group × time ^n^				1.62 ^h^				0.25 ^h^				0.24 ^h^	4.41 ^j^			
				0.001 ***				0.31				0.16	0.001 ***			
Group × HRV × Cognitive function ^o^				1.41 ^h^				0.31 ^h^				0.16 ^h^				
				0.001 ***				0.20				0.34				

^a^ High-frequency power, reflecting PSNS activity; ^b^ Normalized low-frequency power, reflecting SNS activity; ^c^ Low-frequency/high-frequency ratio, reflecting a sympathovagal balance; ^d^ Mini-Mental State Examination; ^e^ Exp: treatment group (lying in bed for 1–7 h); ^f^ Con: control group (lying in bed for 1–7 h); ^g^ Diff: differences between the treatment and control groups; ^h^ Roy’s largest root (group × time interaction); ^i^ Independent sample *t* test to assess the baseline within-group differences; ^j^ Wald χ^2^: generalized estimating equation [33] interaction (group × time point); ^k^ Week 0: baseline; ^l^ Week 5: posttest 1; ^m^ Group: differences between the treatment and control groups; ^n^ Group × time interaction; ^o^ Group × HRV × Cognitive function; *** Indicates *p* < 0.001; HF, high-frequency power; LF%, normalized low-frequency; LF/HF, low-frequency/high-frequency ratio; PSNS, parasympathetic nervous system; SNS, sympathetic nervous system; HRV, heart rate variability; MMSE, Mini-Mental State Examination; SE, standard error.

## Data Availability

Data are accessible upon reasonable request from the respective author.

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
