# Peer review of "Bright Morning Lighting Enhancing Parasympathetic Activity at Night: A Pilot Study on Elderly Female Patients with Dementia without a Pacemaker"

_healthcare, 2023, doi:10.3390/healthcare11060793_

Round 1

Reviewer 1 Report (Previous Reviewer 2)

Paper needs a section that describes the sequence of procedures administered to participants. It appears most of the procedures are described, but the descriptions are presented in different sections of the paper so it is hard to envision the entire sequence.

‘Compared with the treatment group, the 222 control group demonstrated a greater reduction in HF power at Post-test 1 than at base- 223 line (Figure 3b).’ However, Figure 3a shows no change between baseline and post-test 1 in controls. Please explain discrepancy.

The captions for Figures 4a and 4b are reversed.

“The effects of BML may indicate the possible integration of recent work regarding 389 alleviating the allostatic load, polyvagal theory, neurovisceral integration model, and 390 emerging evidence on the roles of gamma-aminobutyric acid (GABA) and decreased 391 glutamate [42]” This sentence is incomprehensible.

“The vagus nerve acts as a “brake” for the SNS when the PSNS deficit 411 leads to an unrestrained SNS [44], as in the washout BML intervention, reducing vagus 412 nerve actions and triggering boosts in the SNS.” What is the washout BML intervention in this study? Were there analyses of the washout BML intervention.

Author Response

  1. Paper needs a section that describes the sequence of procedures administered to participants. It appears most of the procedures are described, but the descriptions are presented in different sections of the paper so it is hard to envision the entire sequence.

Response: Thank you for your suggestion. We have revised the text to ensure that the procedures are described in more detail and with greater consistency.

  1. ‘Compared with the treatment group, the 222 control group demonstrated a greater reduction in HF power at Post-test 1 than at base- 223 line (Figure 3b).’ However, Figure 3a shows no change between baseline and post-test 1 in controls. Please explain discrepancy.

Response: Thank you for your suggestion. We have revised the text to clarify the outcome as follows: “The BML group demonstrated a sustained peak in HF power while lying in bed for 2-4 h. Specifically, the BML group had an increased peak in HF power at 2 h at posttest 1 (Figure 3b). Therefore, the BML group exhibited higher PSNS activity than the control group, resulting in stable and regular circadian rhythms” (please see p. 7).

  1. The captions for Figures 4a and 4b are reversed.

Response: Thank you for your suggestion. We have corrected the captions in the revised manuscript.

  1. “The effects of BML may indicate the possible integration of recent work regarding 389 alleviating the allostatic load, polyvagal theory, neurovisceral integration model, and 390 emerging evidence on the roles of gamma-aminobutyric acid (GABA) and decreased 391 glutamate [42]” This sentence is incomprehensible.

 Response: Thank you for your suggestion. We have revised the text to clarify this sentence as follows: “The effects of BML might include the alleviation of autonomic nervous load, regulation of circadian rhythms, and integration of neuroendocrine systems, the last of which is indicated by an increase in gamma-aminobutyric acid (GABA) levels and decrease in glutamate levels [42]” (please see p. 13).

  1. “The vagus nerve acts as a “brake” for the SNS when the PSNS deficit 411 leads to an unrestrained SNS [44], as in the washout BML intervention, reducing vagus 412 nerve actions and triggering boosts in the SNS.” What is the washout BML intervention in this study? Were there analyses of the washout BML intervention.

 Response: Thank you for this question. Due to the high dropout rate, the analyses were removed from the current version of the manuscript as suggested in a prior review. However, in the literature, a washout BML intervention was proposed to reduce vagus nerve actions and trigger boosts in the SNS.

Reviewer 2 Report (Previous Reviewer 3)

Revision is satisfactory

Author Response

Thank you for your suggestion. 

This manuscript is a resubmission of an earlier submission. The following is a list of the peer review reports and author responses from that submission.

Round 1

Reviewer 1 Report

This manuscript addresses an interesting topic. Generally it  is well written. The background is clearly presented in the Introducition. Methods are timely and appropriate. Novel results are provided in text and tables. Most of the Discussion is appropriate.

However, some corrections and clarifications are necessary.

Line (l) 36 - The first part of the firs sentence of the Introduction "Dementia is... " is difficult to understand. Please rephrase.

l 306 - "We recommend BML for driving sleep propensity." - Please elaborate, which findings support the view. Please give references.

l 311. - "night light kills". Please elaborate, which studies support this statement. The cited reference 31 from "Bussiness Weekly" does not appear to be related to a peer-reviewes medical study.

l 339. "BML therapy is the forefront treatment..." - This view appear exaggerated. Please give supporting references. Are there  guidelines for therapy/prevention of dementia,  neurodegenerative diseases and /or sleep disorders which recommend BML?

L 417- Are there longterm studies supporting the view, that continuous BML is helpful and necessary?

Author Response

  1. Reviewer #1 : .The first part of the firs sentence of the Introduction "Dementia is... " is difficult to understand. Please rephrase.

Response:

Thank you for your suggestion. We have revised the title and text to clarify the introduction (Please see p. 1).

  1. We recommend BML for driving sleep propensity." - Please elaborate, which findings support the view. Please give references.

Response:

Thank you for your comments. We have elaborated this point further and added references (Please see p. 12).

BML exposure can suppress melatonin secretion and induce serotonin secretion, which reduces phase delay and night awakening, thereby reducing sleep disturbance. Relevant studies have reported that BML improves sleep, circadian rhythms, cognitive function, and behavioral and psychological symptoms of dementia [10,13-16]. We recommend BML for driving sleep propensity.

  1. "night light kills". Please elaborate, which studies support this statement. The cited reference 31 from "Bussiness Weekly" does not appear to be related to a peer-reviewes medical study.

Response:

Thank you for your comments. As suggested, we have elaborated this point and added references.

A previous study reported that morning light cures and night light kills[31]. Exposure to light at night (LAN), especially intensive blue enriched white light greatly suppresses melatonin secretion and disrupts sleep homeostasis, causing sleep disturbance, metabolic syndrome, autonomic circadian rhythm disruptions, dyshomeostasis, and neuropsychiatric symptoms [References (Please see p. 13.].

  1. BML therapy is the forefront treatment..." - This view appear exaggerated. Please give supporting references. Are there guidelines for therapy/prevention of dementia,  neurodegenerative diseases and /or sleep disorders which recommend

Response:

Thank you for your comments. We have revised the “forefront treatment” to reflect your suggestion.

For BML therapy to be effective, a threshold of ≥ 2,500 lux light intensity is suggested for at least 60 min/day in the morning for at least 4 weeks [11,15] (Please see p. 13).

  1.  Are there longterm studies supporting the view, that continuous BML is helpful and necessary?

Response:

The meta-analyses of BML have been shown it to be effective in improving sleep disturbance and circadian rhythm [13-15]. In contrast, BML exposure washout will compromise the treatment outcomes [38]; therefore, continuous BML is helpful and necessary (Please see p. 13).

Author Response

  1. Reviewer #2 : Unfortunately there are major problems with this paper. It is so poorly written that at times it is incomprehensible. The Figures are poorly annotated so that they are difficult to interpret. The figures are so small that they are almost illegible (figure 3-b in particular). The results are poorly presented.

Response:

Thank you for your suggestion. We have revised Figure 3-b to reflect your suggestion and edited the text for language (Please see p. 8).

  1. It is unclear how participants were assigned to groups. At the top of page 3 the authors say We performed a non-randomized controlled pilot study At the bottom of the page they say we enrolled 34 patients who were consecutively allocated to the treatment (n=17) or control group (n=17). Whatever method they used resulted in highly significant differences in the gender composition of the two groups. There was one male in the treatment group vs six males in the control group. Gender differences in the response to BML were not examined. Gender is a potential major confound.

Response:

Thank you for your comments. We performed a non-randomized controlled pilot study with a convenience sampling design (Please see p.3).

The participants were elderly patients with dementia. Patients were exposed to bright morning light for at least 60 min/day from 9 am to 10 am, Monday through Friday, over eight weeks, amounting to 40 hours of treatment. This study presented difficult challenges due to high variability, as explored by the statistical software that evaluated the experimental and control groups. This exploration concluded that the participants’ distribution was homogeneous. Conversely, a statistically significant difference was found for gender. Most of the participants were females. As a limitation, the comparison group had more male participants than the experimental group, and the sex ratios of the two groups were significantly different. Thus, we could not determine the effects of bright morning light exposure enhancing parasympathetic activity at night in older males with dementia (Please see p. 15).

  1. There were also group differences in educational attainment. 2/17 patients in the treatment group had completed at least senior high school compared to 8/17 controls. The effects of this potential confound was also not examined.

Response:

Thank you for your comments. This study presented difficult challenges due to high variability, as explored by the statistical software that evaluated the experimental and control groups. This exploration concluded that the participants’ distribution was homogeneous. Conversely, a statistically significant difference was found with respect to sex (Please see p. 6).

  1. The retention rates were 59% in the treatment group vs 47% in the controls. I suspect attrition was not random. No data are provided comparing the demographic characteristics of the two groups at post-test 1 vs post-test 2. Given the extensive attrition by post-test 2 I would suggest limiting the paper to just using post-test 1 as the endpoint.

Response:

Thank you for your suggestion. We have addressed your concerns in the revised Table 1. Although the attrition rate of post-test 2 is high, the data from the BML exposure washout (post-test 2) may be used as a reference for future light-therapy studies.

  1. How were the researchers who rated cognitive function blinded about what group patients were in? The analyses that are contained in 3.31 do not address the effect of BML on cognition. The analyses should address both the direct effect of BML on cognition as well as the mediation model, where the effect of BML on cognition is mediated by HF. At what time point were these analyses conducted? MMSE sores at baseline and Post-test 1 should be included in Table1.

Response:

Thank you for your feedback. We have revised Table 1 in the revised manuscript (Please see p. 6-7). BML intervention improves cognition function and data, which has been reported in a prior publication [16]. The verification of BML on cognition mediated by HF has become the direction of future research (Please see p.11).

  1. For continuous variables SD or SE should be included in addition to means in tables 2 and 3.

Response: Thank you for your suggestion. We have revised Table 2. (Please see p.8-9).

  1. Sensitivity analyses should be done to determine whether dementia type or dementia severity are driving the effect of BML on HF.

Response: Thank you for your suggestion. No statistically significant differences were observed in the demographic characteristics (dementia type and dementia severity at baseline or post-test 1 and post-test 2 between the experimental and comparison groups (Please see p.6).

Reviewer 3 Report

This study examines effect of BML on SNS and PSNS and the consequent effect on cognitive function in old age with various degree of dementia.

Why should the authors use non-randomized study instead of randomized?

In Table 1, only those who did not withdraw and stayed throughout the entire study should be analysed, therefore Treatment n=10 and Control n=8. In this case, sample size appears too small. If authors recalculate using the data for those who stayed until the end of the study, would it affect statistical significance of outcome?

Should there be error bars in Fig.4b and 5b?

Fig.6: any significant difference between mild and moderate cases in case of BML?

Overall I find the Figure presentations and statistic description must be clarified and improved for the results to be more convincing.

Author Response

  1. Why should the authors use non-randomized study instead of randomized?

Response:

This study cannot achieve random sampling. It is a great challenge to accept the subjects randomly due to several technical problems, such as (i) transporting subjects from the community or institution to the light laboratory and (ii) family of the participants insisting on accompanying them during the experiment

  1. In Table 1, only those who did not withdraw and stayed throughout the entire study should be analysed, therefore Treatment n=10 and Control n=8. In this case, sample size appears too small. If authors recalculate using the data for those who stayed until the end of the study, would it affect statistical significance of outcome?

Response:

Thank you for your suggestion. We have revised Table 2 to reflect your suggestion (Please see p.8-9).

  1. Should there be error bars in 4b and 5b?

Response:

Thank you for your suggestion. We have revised Figures 4b and 5b the include error bars. (Please see p.8,10,11).

  1. 6: any significant difference between mild and moderate cases in case of BML?

Response:

Thank you for your valuable insight. We have revised Figure 6 accordingly (Please see p11-12).

The bright morning light (BML) exposure enhances parasympathetic nervous system (PSNS) activity at night, which is more effective for severe dementia than that for cases of moderate and mild dementia. The severe dementia group with PSNS activity a mean of 5.02 and 4.55 for the BML group and the general lighting group, respectively. Moderate dementia with a mean of 5.14 and 4.99 for the BML group and the general lighting group, respectively. Mild dementia with a mean of 3.93 and 4.45 for the BML group and the general lighting group, respectively.

  1. Overall I find the Figure presentations and statistic description must be clarified and improved for the results to be more convincing.

Response:

Thank you for your suggestion. We have revised Figure and Table to reflect your advice.

Round 2

Author Response

    1. The authors have not included data on cognitive function in the two groups in Table 1. They simply provide that the two groups differ a post-test in cognitive

    Response: “The primary pre-test/post-test analyses were based on the intention-to-treat (ITT) sample. The retention rates were 89% and 77% in the treatment and control groups, respectively. Table 1 summarizes the between-group comparisons of the demographic characteristics using chi-squared tests. There were no differences in the baseline and posttest characteristics between the participants across the groups. Conversely, we observed a significant between-group difference in the sex ratio, with men comprising 5.9% and 35.3% of the patients in the treatment and control groups, respectively, at baseline and Posttest 1 (P <.05). Furthermore, Mann–Whitney U tests did not detect differences between groups in terms of age, education level, marital status, dementia severity, dementia type, recruitment source, and medication use. Hence, an independent sample t test was used to assess HF power, LF%, the LF/HF ratio and MMSE score at baseline. No significant differences were observed between groups, indicating that the participants’ distribution was homogeneous.” (Please see p. 6).

    1. The authors do not rigorously address the significant confounds of group differences in gender and educational attainment.

    Response: Thank you for your valuable insight. This study was not able to achieve random sampling due to several practical issues, including difficulties in transporting all the subjects from their communities or institution to the light laboratory and interference from some family of the participants insisting on accompanying them during the experiment. Hence, the corresponding statistical result was not significant enough to distinguish the difference between the control and experimental groups in terms of educational attainment. Nevertheless, it was statistically significant in terms of sex (please see p. 6).

    1. They did not even attempt to address my fourth and fifth points. They did not conduct analyses to determine if the attrition was random, or even describe how the ratings were blinded

    Response: Thank you for the comment. This study utilized a single-blind approach. The attrition was random, according to each participant's willingness to continue participating or withdraw (please see p. 3).

    • The retention rates were 59% in the treatment group vs 47% in the controls. I suspect attrition was not random. No data are provided comparing the demographic characteristics of the two groups at post-test 1 vs post-test 2. Given the extensive attrition by post-test 2 I would suggest limiting the paper to just using post-test 1 as the endpoint.

    Response: Thank you for the comment. We have revised Table 1 to reflect your comment in the revised manuscript, and the paper uses only posttest 1 as the endpoint.

    • : How were the researchers who rated cognitive function blinded about what group patients were in?

    Response: Data collectors were blinded to the patients in each group to decrease the Hawthorne effect. The data include the questionnaires and the data were uploaded to the cloud by the wearable devices. The former was collected by the first author and entered after being confirmed by the doctoral supervisor to ensure that it met the academic rigor.

    • The analyses that are contained in 3.31 do not address the effect of BML on cognition. The analyses should address both the direct effect of BML on cognition as well as the mediation model, where the effect of BML on cognition is mediated by HF. At what time point were these analyses conducted?

    Response: We have revised Table 1 in the revised manuscript (please see p.7). The BML intervention improved cognitive function, as reported in a prior publication [16]. The MMSE scores were collected at baseline and in the 5th week. Verification of the effect of BML exposure on cognition mediated by HF power may become a direction of future research (please see p. 15).

    • MMSE sores at baseline and Post-test 1 should be included in Table1.

    Response: We have revised Table 1 in the revised manuscript (please see p. 7).

Reviewer 3 Report

revision satisfactory

Author Response

 Thank you for your valuable insight. I have completed all revisions according to the reviewer's opinion. This research is the first research on the change of bright light on the heart rate of dementia at night. I wonder if you can read my revised manuscript and give me another chance.
